# What Makes it Ok to Set a Fire? Iterative Self-distillation of Contexts and Rationales for Disambiguating Defeasible Social and Moral Situations

**Kavel Rao**$^{♡*}$    **Liwei Jiang**$^{♡♠*}$    **Valentina Pyatkin**$^{♠}$    **Yuling Gu**$^{♠}$
**Niket Tandon**$^{♠}$    **Nouha Dziri**$^{♠}$    **Faeze Brahman**$^{♠}$    **Yejin Choi**$^{♡♠}$

$^{♡}$Paul G. Allen School of Computer Science & Engineering, University of Washington
$^{♠}$Allen Institute for Artificial Intelligence
{kavelrao,lwjiang}@cs.washington.edu

## Abstract

Moral or ethical judgments rely heavily on the specific contexts in which they occur. Understanding varying shades of *defeasible contextualizations* (*i.e.*, additional information that strengthens or attenuates the moral acceptability of an action) is critical to accurately represent the subtlety and intricacy of grounded human moral judgment in real-life scenarios.

We introduce *defeasible moral reasoning*: a task to provide grounded contexts that make an action more or less morally acceptable, along with commonsense rationales that justify the reasoning. To elicit high-quality task data, we take an iterative self-distillation approach that starts from a small amount of unstructured seed knowledge from GPT-3 and then alternates between (1) self-distillation from student models; (2) targeted filtering with a critic model trained by human judgment (to boost validity) and NLI (to boost diversity); (3) self-imitation learning (to amplify the desired data quality). This process yields a student model that produces defeasible contexts with improved *validity*, *diversity*, and *defeasibility*. From this model we distill a high-quality dataset, $\delta$-RULES-OF-THUMB ($\delta$-RoT), of 1.2M entries of contextualizations and rationales for 115K defeasible moral actions rated highly by human annotators 85.9% to 99.8% of the time.[1] Using $\delta$-RoT we obtain a final student model that wins over all intermediate student models by a notable margin.

## 1 Introduction

Moral or social judgments play a vital role in decision-making, influencing how we perceive actions and behaviors daily. However, these judgments are far from fixed; instead, they are highly *context-dependent*. Contexts surrounding a core action can significantly *strengthen* or *weaken* its

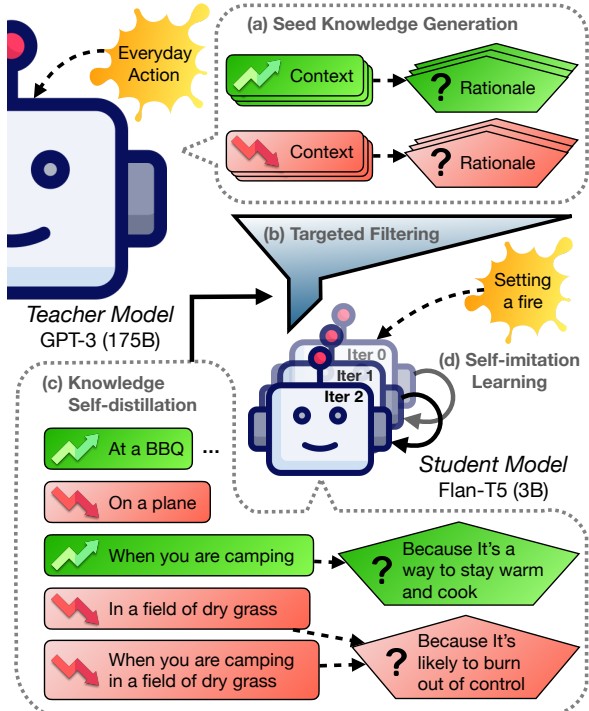

Figure 1: An illustration of the iterative self-distillation pipeline on eliciting *defeasible moral reasoning contextualizations* and *rationales*. For the event "setting a fire," different contextualizations of the action can bend its moral acceptability *up* (*e.g.*, "at a BBQ") or *down* (*e.g.*, "to get revenge"). Capturing the nuances of how additional contexts interplay with base actions is critical for grasping the flexible defeasibility of moral judgments.

moral acceptability. For instance, the act of "knowing where someone lives" may carry no inherent moral weight. But when supplemented by the context that "the purpose is to provide assistance to a person in need," the action becomes more morally justified. Conversely, if the context shifts to "for the purpose of surveillance or spying," the same action loses its moral grounding. This phenomenon of flexibly bending moral rules in instantiations of scenarios is widely recognized in assorted cognitive science studies (Kwon et al., 2022; Levine et al., 2020; Awad et al., 2022; Levine et al., 2018).

The inherent context dependence of moral judg-

---

$^{*}$ Equal contribution.
[1]Dataset is publically available at https://huggingface.co/datasets/kavelrao/d-Rules-of-Thumb

ments underscores the importance of understanding the complex interplay between actions and their grounded contexts in real-life scenarios. Delving into how different contexts bend the moral acceptability of an action, along with the reasons behind these shifts, enables us to make informed moral judgments geared toward situational nuances.

Previous works about contextualized moral judgment pose several challenges. First, they focus primarily on atomic contexts with limited situational complexity. For instance, Ziems et al. (2023) rigidly prescribe grounded contexts to fall under concepts such as settings, roles, and behaviors, narrowing and fragmenting the scope of contextualization. Pyatkin et al. (2023) propose a stepwise clarification question generation system to elicit elementary contexts of moral actions. Another limitation lies in the emphasis on the defeasibility of assumed moral and social judgments (*e.g.*, "it's wrong to yell to your friend"), rather than the natural defeasibility of moral scenarios themselves (Rudinger et al., 2020). Finally, existing works lack the rationales to explain why a particular context renders a situation more or less morally acceptable. We address all the above limitations in this work.

We introduce *defeasible moral reasoning*, a task to provide grounded *contextualizations* (or contexts) that alter the moral acceptability of action, accompanied by commonsense *rationales* that justify the reasoning. We aim to explicitly state underspecified contexts of actions, providing nuance and interpretability to moral judgments. To substantiate this task, we introduce $\delta$-RULES-OF-THUMB ($\delta$-RoT), a high-quality dataset of 1.2M entries of combined contextualizations and rationales for 115K defeasible moral actions. $\delta$-RoT is created through an iterative self-distillation approach. Starting with a small amount of unstructured seed knowledge from GPT-3, we alternate between (1) **self-distillation from student models** to move away from the reliance on expensive API calls for GPT-3; (2) **targeted filtering** using a critic model trained with human judgment to enhance generation validity and natural language inference (NLI) to enhance diversity; and (3) **self-imitation learning** to magnify the desired model properties.

The iterative self-distillation process yields a student model that generates defeasible contexts with enhanced *validity*, *diversity*, and *defeasibility*. From the best-performing student model, we distill the final dataset, $\delta$-RoT. These contextualizations

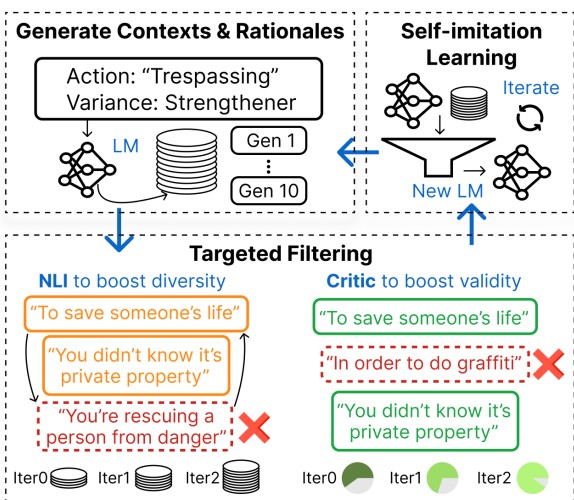

Figure 2: Iterative self-distillation that repeats generation, filtering, and self-imitation learning.

and rationales have been rated highly by human annotators on both validity (85.9%) and language quality (99.8%), ensuring their reliability in capturing the complexities of moral judgment within various contexts. Using $\delta$-RoT, we further train a final downstream student model that prevailed over all intermediate student models.

In sum, in this work, we introduce the defeasible moral reasoning task that involves contexts and rationales for making defeasible moral judgments. We present the iterative self-distillation methodology to gather high-quality and diverse training data for student models (§3) along with careful ablations of each component involved in the pipeline (§5). We distill a sizeable, human-verified dataset under the defeasible moral reasoning task formulation (§2) and a high-performing downstream task model. We will release all student models and $\delta$-RULES-OF-THUMB, along with a subset of human-annotated gold data for training a supervised critic model that mimics human ratings of the validity of contextualizations.

## 2 $\delta$-RULES-OF-THUMB: Dataset Design

We introduce $\delta$-RULES-OF-THUMB ($\delta$-RoT), a dataset for the *defeasible moral reasoning* task. Given an everyday action with a default commonsense moral judgment, $\delta$-RoT captures nuances of the defeasible moral action through contextualizations that either strengthen or attenuate the acceptability of an action. It also contains rationales that explain why the contextualizations affect the judgment. See example data in Figure 1.

As shown in Table 1, $\delta$-RoT contains 115K ac-

tions and 578K entries each of contextualizations and rationales. Extensive human evaluation confirms that δ-RULES-OF-THUMB is of high quality, demonstrating high *validity* (85.9% for contexts; 98.5% for rationales) and *language quality* reflected by fluency and grammar correctness (99.8% for contexts; 99.7% for rationales), on par with human written datasets (West et al., 2022).

**Action and Commonsense Moral Judgment.** We source our scenarios from SOCIAL-CHEM-101 (Forbes et al., 2020), a bank of rules-of-thumb (RoTs) that describe various social, cultural, and moral norms. Each RoT consists of an action and a sociomoral judgment annotated by workers based on natural language snippets of real-life situations.



**Example RoT** (judgment, action):
"It's dangerous to set a fire"



Because RoTs combine everyday atomic actions (*e.g.*, "set a fire") with commonsense sociomoral judgments (*e.g.*, "it's dangerous"), they serve as ideal seeds to be expanded with contextual nuances.

**Morally Variant Contextualization.** *Moral variance* is a binary label such that *strengthening* contextualizations further ground the original action to be more morally acceptable, while *weakening* contextualizations have the opposite effect. Note that meaningful morally variant contextualizations range from simple properties such as locations (*e.g.*, "in a field of dry grass") or auxiliary actions (*e.g.*, "when you're camping") to complex compositional contexts with an intricate interplay between multiple atomic variations such as "when you're camping in a field of dry grass." Thus, we focus on eliciting flexible contexts that exercise concrete and natural effects *tailored* to given actions, instead of pre-defining the categories of the contextualizations regardless of situational nuances (Ziems et al., 2023).

**Commonsense Rationales.** A critical missing piece from previous works on grounded moral judgments is rationale that ties together the actions and contextualizations by explaining the reasoning behind the defeasible effect (*e.g.*, the context "in a field of dry grass" might make the action "setting a fire" less morally acceptable "because it's likely to burn out of control"). δ-RULES-OF-THUMB provides a complete picture of *how* each context achieves a moral variance, paving the way toward dissecting and understanding the varying shades of moral judgments.

| Type | Pol. | Statistics | | Human Val. | |
|---|---|---|---|---|---|
| | | **#Entry** | **#3-Grams** | **%Vld.** | **%Lan.** |
| Action | - | 115K | 110K | - | - |
| Context | All | 578K | 182K | 85.9 | 99.8 |
| | Stren. | 266K | 108K | 84.2 | 100 |
| | Weak. | 312K | 130K | 87.6 | 99.7 |
| Rationale | All | 578K | 275K | 98.5 | 99.7 |
| | Stren. | 266K | 170K | 98.6 | 99.6 |
| | Weak. | 312K | 182K | 98.4 | 99.8 |

Table 1: Statistics and human validation results of δ-RULES-OF-THUMB. **#Entry** is the total number of data entries, and **#3-Grams** is the number of unique 3-Grams of each data type. **%Vld.** is the percentage of valid data rated by humans, and **%Lan.** is the percentage with proper language form (*i.e.*, fluency and grammar).

**Human Critic Gold Data.** In addition to δ-RoT, we also release a dataset of human-annotated quality assessments of machine-generated contextualizations and rationales used to train the critic model for distillation filtering (§3.1). Actions are sampled from SOCIAL-CHEM-101, and we use GPT-3 to generate contextualizations and rationales for both moral variances, each annotated by three crowdworkers. Labels are obtained by majority votes across annotators, but we keep only the subset with full agreement for validation and test sets to ensure high confidence in their labels. The critic gold data contains 11K actions and 20K contextualizations with quality labels across all splits.

## 3 Dataset Creation via Iterative Self-distillation

Competent large language models has opened up new opportunities for automatic dataset creation through symbolic knowledge distillation (West et al., 2022). In this framework, knowledge is generated from a large teacher model, filtered to improve data quality, and then instilled into a smaller student model. Previous works have found that machine-generated datasets can surpass human-authored datasets in their quality and diversity while also achieving greater scale (West et al., 2022; Bhagavatula et al., 2023; Sclar et al., 2022; Jung et al., 2023; Wang et al., 2023).

In this work, we create δ-RULES-OF-THUMB with an iterative self-distillation approach which minimizes the resource bottleneck from expensive GPT-3 API calls. Our approach follows three stages after producing an initial student model using relatively small-scale seed knowledge from GPT-3: (1) **self-distillation from student mod-**

| Model | #Trn. | Top 1 Greedy | | | | | | Top 10 Sampling | | | |
|---|---|---|---|---|---|---|---|---|---|---|---|
| | | **Auto (Critic)** | | **Human** | | | | **Auto (Critic)** | | **Human** | |
| | | Vld. | Avg. | Vld. | Defease. | Lan. | Rationale. | #Vld. | #Unq. Vld. | #Vld. | #Unq. Vld. |
| GPT-3 | - | 0.53 | 0.69 | 0.56 | 0.37 | 0.98 | 0.93 | - | - | - | - |
| **Distill**$_{base}$ | 85K | 0.68 | 0.75 | 0.54 | 0.42 | 0.98 | 0.91 | 6.40 | 5.63 | 5.36 | 4.78 |
| - No Critic | 143K | 0.60 | 0.68 | 0.51 | 0.39 | 0.98 | 0.94 | 5.63 | 5.00 | 4.66 | 4.23 |
| **SelfDistill**$_1$ | 434K | 0.75 | 0.80 | 0.60 | 0.48 | 0.97 | 0.93 | 7.08 | 5.83 | 6.04 | 5.05 |
| - Top 1 Only | 53K | 0.71 | 0.77 | 0.59 | 0.48 | 0.98 | 0.93 | 7.06 | 3.16 | 5.74 | 2.54 |
| - No NLI | 492K | 0.75 | 0.80 | 0.64 | 0.50 | 0.98 | 0.92 | 7.12 | 5.89 | 5.93 | 4.97 |
| **SelfDistill**$_2$ | 466K | 0.79 | 0.83 | 0.62 | 0.50 | 0.98 | 0.93 | 7.60 | 6.15 | 6.26 | 5.21 |
| - Top 1 Only | 57K | 0.73 | 0.78 | 0.62 | 0.49 | 0.97 | 0.93 | 7.28 | 2.60 | 6.13 | 2.16 |
| - No NLI | 567K | 0.80 | 0.83 | 0.63 | 0.51 | 0.98 | 0.91 | 7.65 | 5.92 | 5.93 | 4.73 |
| - No Self-distill | 869K | 0.75 | 0.80 | 0.62 | 0.50 | 0.98 | 0.94 | 7.19 | 5.95 | 6.01 | 5.08 |
| **Distill**$_{final}$ | 578K | **0.86** | **0.88** | **0.71** | **0.56** | **0.99** | 0.92 | **8.40** | **6.45** | **7.26** | **5.69** |

Table 2: Automatic and human evaluation of distilled models across three iterations. We evaluate both the top 1 model generation by greedy decoding and the top 10 candidates by nucleus sampling. Best results are **bolded** and second best results are underlined (to declutter the table, we remove the styles for Lan. and Rationale. as their results are approximately the same across all models).

**els** to move away from reliance on the expensive GPT-3 model; (2) **targeted filtering** to critically select high-quality and diverse data; and (3) **self-imitation learning** to amplify learning signals.

### 3.1 Gathering Medium-quality Seed Data to Train an Initial Student Model

**Eliciting Raw Seed Data from GPT-3.** To set the stage for the later knowledge amplification process via iterative self-distillation, we gather seed knowledge from GPT-3 (175B)[2], the teacher model, to instill into Flan-T5 (3B), a smaller base student model. To do so, we jointly generate defeasible contextualizations for moral actions and associated rationales with carefully engineered task-directed prompts.[3] To encourage diverse alternatives of this initial seed, we generate two contexts/rationales each for both strengthening and weakening moral variances; in total, we obtain 212K contextualizations and rationales for 60K base actions.

**Filtering Raw Seed Data with Critic Model.** Despite careful prompting following the task formulation, raw generations from GPT-3 remain noisy. Thus, we train a binary classifier to simulate human quality assessments on GPT-3 generated contextualizations as inspired by West et al. (2022).[4] To train the critic model, we use the hu-

man quality-assessment gold labels introduced in §2. We fine-tune DeBERTa-V3 (He et al., 2021) on these annotations, resulting in a critic model that achieves high accuracy on a held-out validation set.[5] Using the trained critic model, we filter the teacher model generations to remove errors from the distillation data and obtain an initial medium-quality training corpus, $D_0$, of 85K examples.

**Training Initial Student Model** Our goal is to train an initial student model capable of generating contextualizations and rationales for a given action and moral variance. We fine-tune Flan-T5 (3B) (Chung et al., 2022) on $D_0$ for 3 epochs to produce **Distill**$_{base}$, our base student model.

### 3.2 Refining Intermediate Student Models via Iterative Self-distillation

We refine the quality of the base student model by further amplifying desired generation properties through an iterative self-distillation process, utilizing no additional generations from the teacher model. Our iterative process has some key differences from Bhagavatula et al. (2023), in that we focus on improving *diversity* in addition to quality.

**Self-distillation from Student Models.** First, we generate a corpus of contextualizations and rationales using **Distill**$_{base}$ on a set of newly sampled actions from the training split of SOCIAL-CHEM-101. Given an action, we use nucleus sampling (Holtzman et al., 2020) ($p = 0.9$) to produce 10 contextualizations and rationales for each moral variance.

---

[2] *text-davinci-003* is used wherever GPT-3 is mentioned

[3] See Appendix A for GPT-3 prompt details.

[4] Preliminary human evaluation results show that whenever contextualization is deemed high quality, the rationale is most likely to be high quality too (over 90% of the time). Therefore, although some improvements could be gained on the rationales, in this work, we focus on improving the quality of contexts which starts at ~50% valid, as there's much more room to improve their quality.

[5] See critic model training details in Appendix D.1

**Targeted Filtering.** Next, we again perform targeted filtering on the newly self-distilled data to (1) ensure the validity of the data via the supervised critic model, similarly to the treatment of $D_0$ described in §3.1; (2) encourage diverse model outputs by reducing repetition among valid contextualizations using a Natural Language Inference filter (NLI) (Liu et al., 2022).

NLI is an NLP task that determines whether a premise statement entails or implies the truth of a hypothesis statement (Bowman et al., 2015; Liu et al., 2022). For a given pair of contextualizations $A$ and $B$, we say the pair is mutually entailed if both $A \rightarrow B$ and $B \rightarrow A$ are entailments, indicating a high confidence of not only lexically but also semantically repetitive content. We filter out these mutual entailments such that at most one example from each pair remains in the dataset, thereby removing redundant signals in the training data. We use RoBERTa-Large pretrained on the WANLI dataset (Liu et al., 2022) to compute entailment scores between each of the generated contextualizations for a given input. Formally, the filtering process is defined as:

$$\text{accept}_{NLI}(c_i) = \forall j \in [1, i) : \neg \text{accept}_{NLI}(c_j) \vee$$
$$(P_{NLI}(c_i, c_j) < 0.5 \vee P_{NLI}(c_j, c_i) < 0.5)$$

where $\text{accept}_{NLI}(c)$ determines if a context $c$ is accepted into the filtered set of unique candidates, $c_k$ is the $k$'th candidate, and $P_{NLI}(c_1, c_2)$ is the predicted score that context $c_1$ entails $c_2$.

This process results in a filtered self-generated corpus, $D_1$, of 434K examples. We then train **SelfDistill$_1$** using $D_1$ starting from **Distill$_{base}$**.

To improve the student model further, we repeat the self-distillation process using **SelfDistill$_1$** as the base. Using **SelfDistill$_1$**, we generate a high-quality corpus, $D_2$, automatically filtered by the supervised critic model and NLI to train a second-iteration self-distilled student model, **SelfDistill$_2$**.

### 3.3 Training a Final Student Model with Large-scale, High-quality Data from Refined Self-distilled Model

Using **SelfDistill$_2$**, we produce $\delta$-RULES-OF-THUMB by generating contextualizations and rationales for 94K actions and combining them with previous training sets. We apply the NLI filter as above and filter by a restrictive critic model threshold of 0.96 to ensure high confidence of quality in the dataset. Human evaluation of a subset of 1000 samples shows that 85.9% of contextualizations

| Model | Vld. | Avg. |
|---|---|---|
| GPT-3 (Teacher) | 0.53 | 0.69 |
| Falcon-7B-Instruct | 0.39 | 0.54 |
| GPT-3.5 (ChatGPT) | 0.71 | 0.77 |
| GPT-4 | 0.77 | 0.82 |
| **Distill$_{final}$** | **0.86** | **0.88** |

Table 3: Our **Distill$_{final}$** model outperforms all other baseline models on the top 1 generation via the automatic evaluation.

and 98.5% of rationales are deemed high quality (see details of dataset stats and human validation in Table 1). Using this large-scale, high-quality dataset, we train a final student model, **Distill$_{final}$**, outperforming all previous intermediate student models and the teacher model (*i.e.*, GPT-3).

## 4 Experimentation Setups

### 4.1 Evaluation Data

We hold out a test set of 6K actions from the same distribution in SOCIAL-CHEM-101. For each model we generate contextualizations and rationales over the test set using both greedy decoding (top 1 candidate) and nucleus sampling (top 10 candidates) with $p = 0.9$.

### 4.2 Evaluation Metrics

We aim to evaluate the overall quality of model-generated contextualizations and rationales (*validity*), a model's capability to produce diverse contextualizations for a given action and moral variance (*diversity*), and also the degree of moral variance that the model's contextualizations provide (*defeasibility*). Finally, we also evaluate the general *language quality* of model generations reflected by fluency and grammatical correctness.

**Validity.** For contextualizations, we use the critic model for automatic evaluation. For greedy generations, we compute the ratio of generations in the test set that pass the critic filter threshold as defined in §3 (**Auto/Vld.**) and the average predicted critic score across the test set (**Auto/Avg.**). For sampled generations, we compute the average number of contextualizations out of the top 10 candidates that pass the critic filter threshold (**Auto/#Vld.**). We also conduct human evaluation to vet validity of contextualizations to complement conclusions drawn from automatic evaluation (**Human/Vld., #Vld.**). We evaluate the validity of rationales with human evaluation (**Rationale.**).

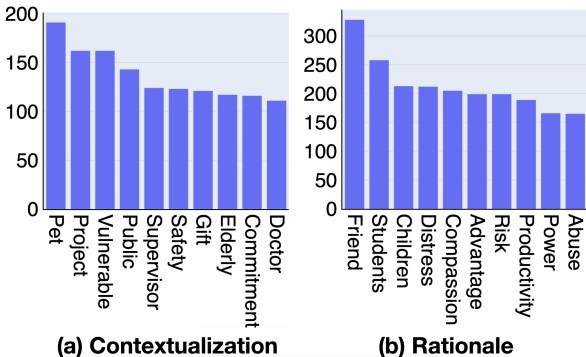

**(a) Contextualization**    **(b) Rationale**

Figure 3: Top 10 topics and their counts among 10K sampled contextualizations and rationales from $\delta$-RoT.

**Diversity.** We use automatic evaluation to assess the diversity of generated contextualizations as it is a generally well-defined dimension. Similarly to the mutual entailment NLI filter in §3, we compute the bidirectional entailment probabilities between each pair of contextualizations across valid candidates and report the average number of semantically unique generations (**Auto/#Unq. Vld.**). This metric describes the model's capability to produce multiple varied contextualizations for a given input, directly indicating the diversity of model outputs.

**Defeasibility.** We break down human evaluation of contextualization validity into more granular answer choices—"significantly" or "slightly" shifting the moral implication of the base action. Defeasibility of the contextualization (**Defease.**) is computed as $(\#contexts_{significant}*1+\#contexts_{slight}*0.5)/\#all$, indicating the degree to which the contextualizations affect the morality of the original actions. See Appendix B for annotation details.

**Language Quality.** We evaluate the language quality (*i.e.*, fluency and grammar correctness) of generated contexulizations and rationales with human evaluations (**Lan.**).

## 5 Results and Discussions

In this section, we present results and insights of the iterative self-distillation process and an analysis of the resulting dataset, $\delta$-RoT.

### 5.1 Insights of Iterative Self-distillation

As shown in Table 2, student models improve across iterations during the iterative self-distillation process on all of *validity* (0.54→0.71), *diversity* (4.78→5.69), and *defeasibility* (0.42→0.56). In particular, the final student model, **Distill**$_{final}$, wins over GPT-3 (the teacher model orders of magnitude larger) by a substantial relative gain on va-

lidity (26.8%) and defeasibility (51.4%), demonstrating the effectiveness of distilling small specialized knowledge models from large general-purpose close-sourced models like GPT-3.

**Filtering by the critic model improves the quality of contextualizations.** Our results show that filtering training examples by the critic model improves the quality of generated contextualizations, in line with previous findings (West et al., 2022; Bhagavatula et al., 2023). In particular, we conduct an ablation study without using critic filtering (**Distill**$_{base}$-*No critic*), resulting in lower performance on almost all contextualization metrics and similar performance on others, despite its training set being ∼70% larger than **Distill**$_{base}$.

**Training student models on diverse self-generated data improves validity and diversity over greedy decoding.** We find that omitting diverse candidates during the self-distillation process results in a drastic decrease in the diversity of the subsequent models. In particular, ablations using only the top 1 candidate in distillation (**SelfDistill**$_1$-*Top 1 Only* and **SelfDistill**$_2$-*Top 1 Only*) produce significantly less valid and unique generations compared to **SelfDistill**$_1$ (5.05→2.54) and **SelfDistill**$_2$ (5.21→2.16). This insight is critical as previous symbolic knowledge distillation works (West et al., 2022; Bhagavatula et al., 2023) focused primarily on improving the validity of downstream student models without screening the diversity.

**Filtering repetitions with NLI improves the diversity of candidates from student models.** Is training on more candidates itself, without filtering out repetitions, sufficient for improving the diversity of downstream models? To answer this question, we conduct ablation studies without using the NLI mutual entailment filter, *i.e.*, **SelfDistill**$_1$-*No NLI* and **SelfDistill**$_2$-*No NLI*. Our results show that despite being trained with more data, these two models generate less valid and unique contextualizations compared to **SelfDistill**$_1$ (5.05→4.97) and **SelfDistill**$_2$ (5.21→4.73), shedding light on the importance of having *truly* diverse training data by removing redundancy.

**Successive iterative training leads to a higher quality student model than a single iteration.** We train an ablation model (**SelfDistill**$_2$-No Self-distill) combining the actions in the training sets of both the first and second rounds of distillation,

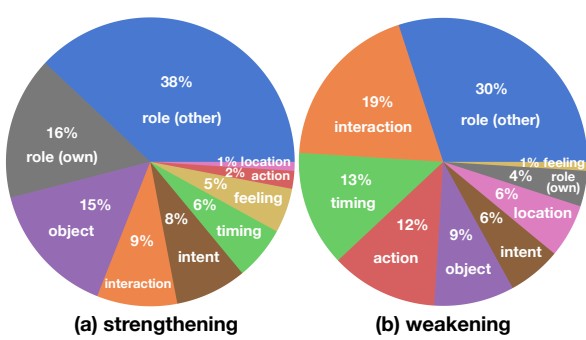

Figure 4: Qualitative analysis of contextualization categories per moral variance. (a) and (b) are for strengthening and weakening contexulizations, respectively.

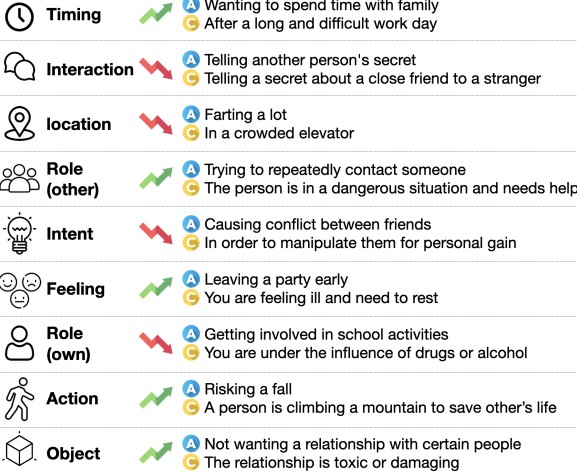

Figure 5: Example rich contextualizations per category.

but with only one iteration of self-learning from **Distill**$_{base}$. **SelfDistill**$_2$, which has been trained using the same actions over two rounds of successive training for the same number of total training steps, outperforms this ablation on almost all metrics, showing the effectiveness of amplifying learning signals via successive iterations of self-learning.

**The final student model outperforms orders of magnitude larger off-the-shelf models.** We evaluate the zero-shot performance of some of the most powerful general-purpose LLMs of varied sizes with 1000 sampled examples from our test set. We use the same instructions as we used to distill the seed knowledge from GPT-3 to prompt these baselines in a zero-shot manner (see Appendix §A). Results in Table 3 show that despite some of the zero-shot models being orders of magnitude larger than our final student model (*e.g.*, 175B vs. 3B), our model outperforms them all, proving the effectiveness of our proposed approach.

### 5.2 Delving into δ-RULES-OF-THUMB

We analyze δ-RULES-OF-THUMB to gauge the dataset composition and gain a comprehensive pic-

ture of captured contextualizations and rationales.

**Contextualization.** To understand what topics contextualizations in δ-RoT represent, we conduct a topic analysis with BERTopic (Grootendorst, 2022), an easily interpretable off-the-shelf topic modeling technique. Frequent topics of contextualizations are shown in Figure 3(a), which involve *daily objects or entities* (*e.g.*, pet, project, supervisor, gift, doctor) and *characters or properties that carry moral weights* (*e.g.*, vulnerable, public, safety, elderly, commitment). In particular, *vulnerability* serves as a critical weakening context if characters in the action are among vulnerable populations (*e.g.*, "taking advantage of people" becomes less acceptable if "they are in a vulnerable state"). See topics analysis details in Appendix §E.1.

We also manually analyze the categories of 200 contextualizations sampled from δ-RoT. As shown in Figure 4, frequent types of contextualizations include role specifications of the action taker (own), other characters involved in the scene (other), and setting specifications such as object, timing, and location. In addition to individual role specifications, interactions, relationships, and dynamics between multiple roles add rich groundings that carry moral significance. Figure 5 shows example contextualizations under each category. Contextualizations in δ-RoT have an average of 11.7 words, providing concrete, specific contexts tailored to each action.

We further conduct a qualitative *error analysis* over generations from the **Distill**$_{final}$ to gauge what makes a context implausible or incorrect.

**Trivial Context:** Context that adds details that may often be relevant to morality for other actions, but is rendered trivial in this particular case, *e.g.*, "Staying up all night" vs. "Staying up all night while in a relationship."

**Infeasible/Unlikely/Unnatural Context:** Context that is infeasible, highly unlikely in a real world setting, or unnatural by itself and/or in relation with the action, *e.g.*, "Participating in a chess team" vs. "The team is made up of people who have been convicted of a serious crime."

**Opposite Context:** Context that adds details opposite to the desired moral variance, *e.g.*, "Offering people money" vs. "Offering money to someone who is in a vulnerable financial situation" when prompted for a weakening context.

**Rationales.** We conduct the same topic analysis on rationales. Results in Figure 3(b) highlight

that common topics in justifying a moral decision involve *important roles* (*e.g.*, friend, students, children) and *common human values* (*e.g.*, distress, compassion, risk, productivity, power, abuse). See topics analysis details in Appendix §E.1.

Diving into specific examples of why contexts shift the moral implications of actions, we find common values that uplift the acceptability of an action include *empathy*, *kindness*, *support*, and *respect* (*e.g.*, "...someone who is in need of emotional support, which shows empathy and kindness"). Additionally, *(in)equality* or *(un)fairness* is another dimension of value that carries significant weight on moral implications (*e.g.*, "...taking advantage of someone else's generosity when you have the resources to provide for yourself"). Finally, contexts that are explained to promote or impede *physical/mental wellbeing*, *financial health*, or *learning/working productivity* (*e.g.*, "...helping them learn, which is beneficial for their future") are also common. These qualitative results show consistency with the automatic topic analysis.

**Toxicity Analysis.** Since the seed actions of $\delta$-RoT sourced from SOCIAL-CHEM-101 (Forbes et al., 2020) mainly concern everyday situations, they are at a low risk of containing toxic content. In addition, due to the careful filtering with the critic model and the iteratively refined self-distillation process, we expect most of the low-quality (including potentially biased data points) to be already filtered out from the final dataset. However, because any toxicity in moral reasoning data is especially detrimental and could easily propagate through downstream tasks, we run a toxicity analysis on a subset of 40K data points from $\delta$-RoT using the Perspective API (Lees et al., 2022). Our results show that the average toxicity score is *0.09*, indicating very low toxicity overall. In a qualitative analysis of the data rows with higher toxicity scores (with a max of 0.83), we observe a strong pattern where the base action itself is problematic, and the distilled contexts produce the target moral variance without contributing significantly to the toxicity of the complete statement (see examples in Table 9 of Appendix §E.2). While no existing toxicity detection method can accurately measure all potential biases, this analysis provides reasonable confidence in the lack of toxicity in our generated contextualizations and rationales.

**Cultural Biases** It's also important to note the sensitivity of moral reasoning to cultural differences. In fact, previous studies have pointed out that cultural bias is a pervasive phenomenon across many NLP models (e.g., GPT-3/3.5/4) and tasks (e.g., hate speech detection with Perspective API, RewireAPI, HateRoberta) (Santy et al., 2023). To better represent diverse perspectives to our contextualizations, we (1) abstain from producing an absolute moral judgment given an action and (2) promote diverse distillation as discussed previously.

However, these measures cannot eliminate all traces of bias in the final model, so we also qualitatively probe $\delta$-RoT to examine cultural biases. Admittedly, as our dataset and student models are distilled from GPT-3, which is shown to present *Western-centric* perspectives, it is likely that our dataset and models inherit this cultural bias as well (Santurkar et al., 2023; Abdulhai et al., 2023). For example, when prompted for a weakening contextualization for the action "Not having freedom of speech in {country}." For some countries such as Japan, the United Kingdom, and the United States, the top generated context is "in a workplace setting." Yet for other countries such as China, India, Thailand, Korea, and Russia, **Distill**final produces different results which might imply these countries have varying levels of human rights concerns (see details in Appendix §E.3). This example aligns with our intuition that the student model might display Western-centric biases, and it fits with a previous finding by Fraser et al. (2022) that such models are likely to encode the cultural biases aligned with those involved in training data annotation.

Thus, despite our careful filtering process, it is clear that culturally biased generations can still be produced by our model and may be present in $\delta$-RoT. As such, users of this dataset must exercise discretion and care when applying it to novel use cases, and it should never be used as prescriptive ethical advice. This points to a key direction for future work to further enrich multicultural representations in computational moral reasoning and other commonsense understanding tasks.

## 6   Related Work

**Computational Morality.** Jiang et al. (2022) present Delphi, a commonsense moral model trained to present a descriptive view of ethical judgments. Ammanabrolu et al. (2022); Hendrycks et al. (2021); Pan et al. (2023) incorporate moral values

in an interactive game environment to align agent actions with social norms. Kim et al. (2022) uses social norms to guide conversational agents' prosocial responses. Jin et al. (2022) introduce MoralExceptQA, a task of identifying the acceptability of breaking a well-known moral rule in different situations. Fung et al. (2023) introduce NormSAGE, a framework to discover multi-Lingual and multi-cultural norms on-the-fly. There is also a prominent line of work in quantifying social, political, and moral values and views presented in language models by using well-established public opinion surveys or social science instruments (Santurkar et al., 2023; Hartmann et al., 2023; Fraser et al., 2022). Recently, Sorensen et al. (2023) builds the Kaleido model to capture the importance of pluralistic human values in moral decision-making.

**Defeasible Reasoning.** Defeasibility describes the idea that new information might strengthen or weaken a given interpretation (Rudinger et al., 2020). This concept has been used in multiple works for different applications: Rudinger et al. (2020) introduced two task formulations, one which concerns generating strengthening or weakening updates to a premise and hypothesis, and the other one which concerns classifying whether a premise and an update strengthen or weaken the hypothesis. Madaan et al. (2021) improved upon the latter task by modeling inference graphs. Our work relates to recent efforts towards contextualizing moral reasoning. Pyatkin et al. (2023) developed ClarifyDelphi, a system capable of asking clarification questions to elicit the context surrounding a judgment. With NormBank, Ziems et al. (2023) introduce a framework for grounded reasoning about norms, adding environmental conditions and agent characteristics. Rather than a QA setup or providing an atomic groundings, in $\delta$-RoT, we instead provide free-text contextualizations along with supporting rationales which justify how each piece of context alters the morality of the an action.

**Explanations and Rationales.** Free-form rationales have emerged as a promising direction to promote models' reasoning capabilities and aid interpretability by filling in the knowledge gap. Prior works on rationale generations take either a supervised approach by training on human-written explanations (Camburu et al., 2018; Rajani et al., 2019; Narang et al., 2020; Kumar and Talukdar, 2020) or a weakly supervised approach (Glockner

et al., 2020; Brahman et al., 2021). The advent of in-context learning (Brown et al., 2020; *inter alia*) led to growing interest in using LLMs to generate rationales in few-shot prompting mode (Wiegreffe et al., 2022; Marasovic et al., 2022; Wei et al., 2022). While so-called explanation-based prompting shows encouraging results, it is hindered by costly API calls. We instead endow accessible models with joint generation of contextualizations and rationales, reducing the computation required.

**Automatic Data Generation.** Previous works on automatic data generation have worked on creating datasets for commonsense reasoning (West et al., 2022; Bhagavatula et al., 2023; Liu et al., 2022; Wang et al., 2023; Kim et al., 2023), dialogues (Kim et al., 2023; Xu et al., 2023; Geng et al., 2023; Chiang et al., 2023), and summarization (Sclar et al., 2022; Jung et al., 2023). West et al. (2022) propose the symbolic knowledge distillation framework, with several follow-up works to extend it with iterative distillation (Sclar et al., 2022; Jung et al., 2023; Bhagavatula et al., 2023). We further build on this paradigm to encourage diversity in distillation and apply our method to moral contextualization and rationales.

## 7   Conclusion

In this work, we highlight the importance of dynamic contexts in shaping moral reasoning. We introduce *defeasible moral reasoning*, a task of providing grounded contexts to elucidate varying degrees of moral acceptability, accompanied by commonsense rationales to justify the reasoning.

We employ an iterative self-distillation methodology to create a high-quality and diverse dataset, $\delta$-RULES-OF-THUMB, comprising over 1.2M combined entries of contextualizations and rationales for 116K defeasible moral actions. Through this iterative approach, we also obtain a small student model capable of generating defeasible contexts with improved validity, diversity, and defeasibility.

Our work aims to promote a deeper understanding of the intricate interplay between defeasible moral actions and grounded contexts that shape moral acceptability in a nuanced and complicated way, building a strong foundation for future works on enriching cultural representations in computational moral reasoning research. We hope $\delta$-RoT serves as a rich resource for the community to study how moral judgments are made to unveil this unique perspective of human intelligence.

## Limitations & Ethical Considerations

Large-language models can generate text that might be biased and insensitive to a user's socio-cultural context (Bordia and Bowman, 2019; Sharma et al., 2021; Hovy and Prabhumoye, 2021). By introducing the *defeasible moral reasoning* task, we consider different contexts and rationales, making a step towards being more diverse and inclusive in accounting for different perspectives.

However, even with our filtering by the critic model, it is possible for biased or incorrect outputs to be produced by distilled models. The critic model is trained to be a strong approximation of human judgment, but it is not perfect, and due to the scale we cannot collect human annotations to verify all examples in model training data or $\delta$-RULES-OF-THUMB.

In addition, determining moral variance is a form of moral judgment and so may not have a clear answer in some cases, *e.g.*, trolley problems. There are certainly contextualizations which different groups of people disagree over whether they make the base action more or less acceptable, as could be seen in our critic gold data, where we employed inter-annotator voting to reconcile differing opinions.

With these points in mind, our dataset and models should *never* be used as direct moral advice to humans; they are solely intended to be resources for more nuanced reasoning and interpretation in computational morality research.

We will publicly release our dataset and final student model to promote open-source research but for gated research purposes only. To mitigate risks of misuse, we will inform all users about potential ethical implications and risks of the resource and require them to complete a data/model user agreement form to acknowledge their consent to proper usage of the resource. This will also help us track how our resource is repurposed for downstream applications.

Finally, for extremely morally negative situations, there might not be any reasonable contextualizations to make the action justifiable (*e.g.*, "genocide"). However, the situations we source from SOCIAL-CHEM-101 focus on everyday actions that do not carry extreme moral implications. Therefore, we consider identifying impossible cases to further contextualize out of the scope of our current study. Future work could investigate more inherently morally charged cases that could not be justified even with further contextualizations.

## Acknowledgement

The authors thank the anonymous reviewers. This research was supported in part by DARPA under the ITM program (FA8650-23-C-7316) and the Allen Institute for AI.

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

## A GPT-3 Prompt for Seed Data Generation

We prompt GPT-3 with the action and moral variance to jointly generate the contextualization and rationale in a zero-shot setting. We use nucleus sampling with $p = 0.9$ and presence and frequency penalties of $0.5$.

We explore multiple prompts, including few-shot versus zero-shot prompting and multiple variations of wording. From qualitative analysis, we find that this zero-shot prompt performs well, while also reducing the number of tokens required for each request.

```
Given an action, write down a
situation in which the action is
more {ethical/unethical}
and give a reason for
why it makes the action
more {ethical/unethical}.
Use the following format.

Action:
Situation:
Explanation:

Action: {action}
Situation:
```

| Type | Pol. | %Vld. | Human Val. %Significant. | %Slight |
|------|------|-------|--------------|---------|
| Context | All | 85.9 | 79.5 | 20.5 |
| | Stren. | 84.2 | 79.0 | 21.0 |
| | Weak. | 87.6 | 80.0 | 20.0 |
| Rationale | All | 98.5 | 93.4 | 6.6 |
| | Stren. | 98.6 | 92.9 | 7.1 |
| | Weak. | 98.4 | 93.9 | 6.1 |

Table 4: Expanded human validation results of $\delta$-RULES-OF-THUMB, breaking down the degree of moral variance and logical completeness. **%Vld.** is the same as Table 1. **%Significant** and **%Slight** are (for Context) the percent of **Vld.** which significantly or slightly impact the moral variance relative to the base action; and (for Rationale) the percent of **Vld.** which are annotated as fully or somewhat valid respectively.

## B Human Annotation Details

We use Amazon Mechanical Turk as the interface for all human annotations and evaluations. For each task, we estimate the completion time by doing a selection of jobs ourselves in order to target a compensation rate of $15 per hour.

### B.1 Critic Model Data Collection

For the critic training data, we collect human annotations on the quality of GPT-3 generated contextualizations, which we then portion into an 80%/10%/10% train/validation/test split. We combine the "Neutral" and "Opposite" answer choices into a single "Invalid" label. As described in §2, to reduce noise in the dataset, we collect 3 annotations per generation and vote to produce the gold label; for validation and test sets we include only cases where all three annotators agree.

### B.2 Human Evaluation of Model Generations and the Final Distilled Dataset

We design the dataset human evaluation to gather fine-grained assessments on contextualizations and rationales. As such, we include options for "slightly" valid contextualizations and "somewhat" valid rationales along with "invalid", which allows us to gain a more in-depth understanding of the quality of the data. In Table 1 we collapse these labels into simply "valid" and "invalid", considering the first two options "valid" and only the last option "invalid".

We find high inter-annotator agreement on these evaluations, with the questions on language quality and rationale validity at over 90% full three-

**Sentence:**

**${source-action-0}** is **${modifier-0}**
when **${generation-update-0}**
because **${generation-explanation-0}**

**Q1.** [Update Effectiveness] Does the **update** make the **action** **${modifier-0}**?

- ○ [Yes] It makes the action **${modifier-0}**.
- ○ [Neutral] It does not make the action **${modifier-0}** or the opposite.
- ○ [Opposite] It has the opposite effect of making the action **${modifier-0}**.

**Q2.** [Explanation Effectiveness] Does the **explanation** provide facts or reasoning that explain how the **update** makes the **action** **${modifier-0}**?

**Not applicable:** Update Effectiveness is not marked as [Yes]

---

**Sentence:**

**${source-action-0}** is **${modifier-0}**
when **${generation-update-0}**
because **${generation-explanation-0}**

**Q1.** [Update Effectiveness] Does the **update** make the **action** **${modifier-0}**?

- ⦿ [Yes] It makes the action **${modifier-0}**.
- ○ [Neutral] It does not make the action **${modifier-0}** or the opposite.
- ○ [Opposite] It has the opposite effect of making the action **${modifier-0}**.

**Q2.** [Explanation Effectiveness] Does the **explanation** provide facts or reasoning that explain how the **update** makes the **action** **${modifier-0}**?

- ○ [Yes] It provides **facts or reasoning** that **clearly** explain how the update makes the action **${modifier-0}**.
- ○ [Somewhat] It provides **facts or reasoning** that **somewhat** explain how the update makes the action **${modifier-0}**.
- ○ [No] It does not effectively reason about how the update makes the action **${modifier-0}**.

Figure 6: The human data collection template for the critic model gold training data collection.

annotator agreement. On the context validity question, we find 57% full agreement, and in 82% of cases, two out of three annotators agree on a label. We expect a lower agreement on this question compared to others, since the moral judgment of the action and context is an inherently subjective task, and the answer may not be clearly defined in all cases. With the high two-way agreement, we have confidence in the accuracy of labels after applying voting across annotators.

## C  Full Iterative Self-distillation Algorithm

---

**Algorithm 1** Iterative Self-distillation of $\delta$-RoT

---

**Require:** teacher model $\tau$, critic model $\rho$, $A_{\text{SocialChem}}$
1: $A_0 \leftarrow$ sample $A_{\text{SocialChem}}$
2: $A_{\text{SocialChem}} \leftarrow A_{\text{SocialChem}} \setminus A_0$
3: $D_0 \leftarrow$ GENERATEDIVERSE$(\tau, A_0)$
4: $D_0 \leftarrow$ FILTER$(\rho, D_0, \text{Threshold}_{\text{distill}})$
5: **Distill**$_{base} \leftarrow$ Fine-tune base model on $D_0$
6: **for** $i = 1, 2$ **do**
7:     $A_i \leftarrow$ sample $A_{\text{SocialChem}}$
8:     $A_{\text{SocialChem}} \leftarrow A_{\text{SocialChem}} \setminus A_i$
9:     **SelfDistill**$_i, D_i \leftarrow$ SELFDISTILL$(\sigma, A_i, \rho)$
10: $D_{\text{rem}} \leftarrow$ GENERATEDIVERSE$(\sigma, A_{\text{SocialChem}})$
11: return FILTER$(\rho, D_0 \cup D_1 \cup D_2 \cup D_{\text{rem}}, \text{Threshold}_{\text{dataset}})$
12: **procedure** SELFDISTILL$(\sigma_{\text{old}}, A, \rho)$
13:     $D \leftarrow$ GENERATEDIVERSE$(\sigma, A)$
14:     $D_f \leftarrow$ FILTER$(\rho, D, \text{Threshold}_{\text{distill}})$
15:     $\sigma_{\text{new}} \leftarrow$ Fine-tune $\sigma_{\text{old}}$ on $D_f$
16:     return $\sigma_{\text{new}}, D_f$
17: **procedure** GENERATEDIVERSE$(\mu, A)$
18:     $D \leftarrow \emptyset$
19:     **for** $a$ in $A$ **do**
20:         **for** $p = +, -$ **do**
21:             $b \leftarrow \{$Top 10 beams from $\mu_{\text{old}}$ for $a, p\}$
22:             $D_b \leftarrow \emptyset$
23:             **for** $(a, p, c, r)$ in $b$ **do**
24:                 **if** $\forall c' \in D_b : \neg$MUTUALNLI$(c, c')$ **then**
25:                     $D_b \leftarrow D_b \cup \{(a, p, c, r)\}$
26:             $D \leftarrow D \cup D_b$
27:     return $D$
28: **procedure** FILTER$(\rho, D, \kappa)$ $D_f \leftarrow \emptyset$
29:     **for** $(a, p, c, r)$ in $D$ **do**
30:         **if** $\rho(a, p, c) > \kappa$ **then**
31:             $D_f \leftarrow D_f \cup \{(a, p, c, r)\}$
32:     return $D_f$
33: **procedure** MUTUALNLI$(c1, c2)$
34:     return Entail$(c1, c2) \wedge$ Entail$(c2, c1)$

---

## D  Model Training Details

### D.1  Critic Model

We fine-tune the critic model from DeBERTa-V3-Large (He et al., 2021) on the human critic gold data §2, attaching a 2-layer classificaton head with a hidden size of 512 and GELU (Hendrycks and Gimpel, 2016) as the activation function.

Since the moral variance of a given contextualization may be ambiguous or contested between different annotators, we filter the validation and test data to only the subset on which all annotators agree on a label.

We conduct a small search of hyperparameters using the dev set around the values suggested by West et al. (2022), and we find training with batch size of 4, learning rate of $5e{-}06$, and dropout of 0.1 to be effective to produce a discerning critic model. Because the dataset is heavily class-imbalanced with about 3:1 high-quality to low-quality contextualizations, we weight the loss of low-quality examples in training with the reciprocal of the imbalance. We employ early stopping to save the checkpoint with the lowest validation loss after 15000 training steps. We use the special tokens **[ACTION]** to denote the start of the action, and **[POS]** and **[NEG]** respectively to denote a contextualization with positive and negative moral variance. The critic model training takes approximately 3 hours to train on a single NVIDIA Titan XP GPU.

| | Accuracy | F1 Score | AUC PR Curve |
|---|---|---|---|
| **Val** | 0.88 | 0.93 | 0.98 |
| **Test** | 0.86 | 0.92 | 0.98 |

Table 5: Critic model metrics on evaluation sets from gold human-annotated data (§2)

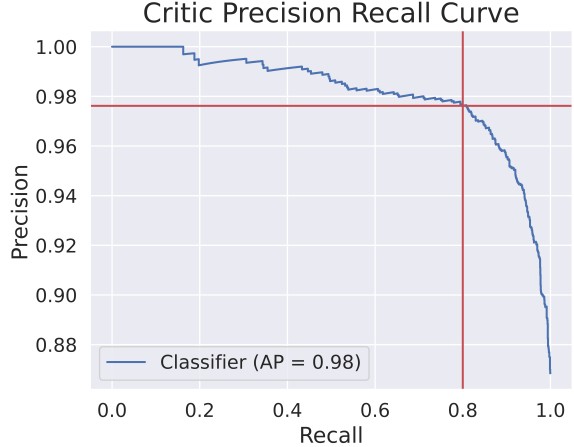

Figure 8: PR Curve on Validation Set. Red lines display recall of 0.8 with high precision used to select threshold

Using the precision-recall curve, we determine a critic threshold of 0.8 for distillation, which achieves high precision and recall on the validation set.

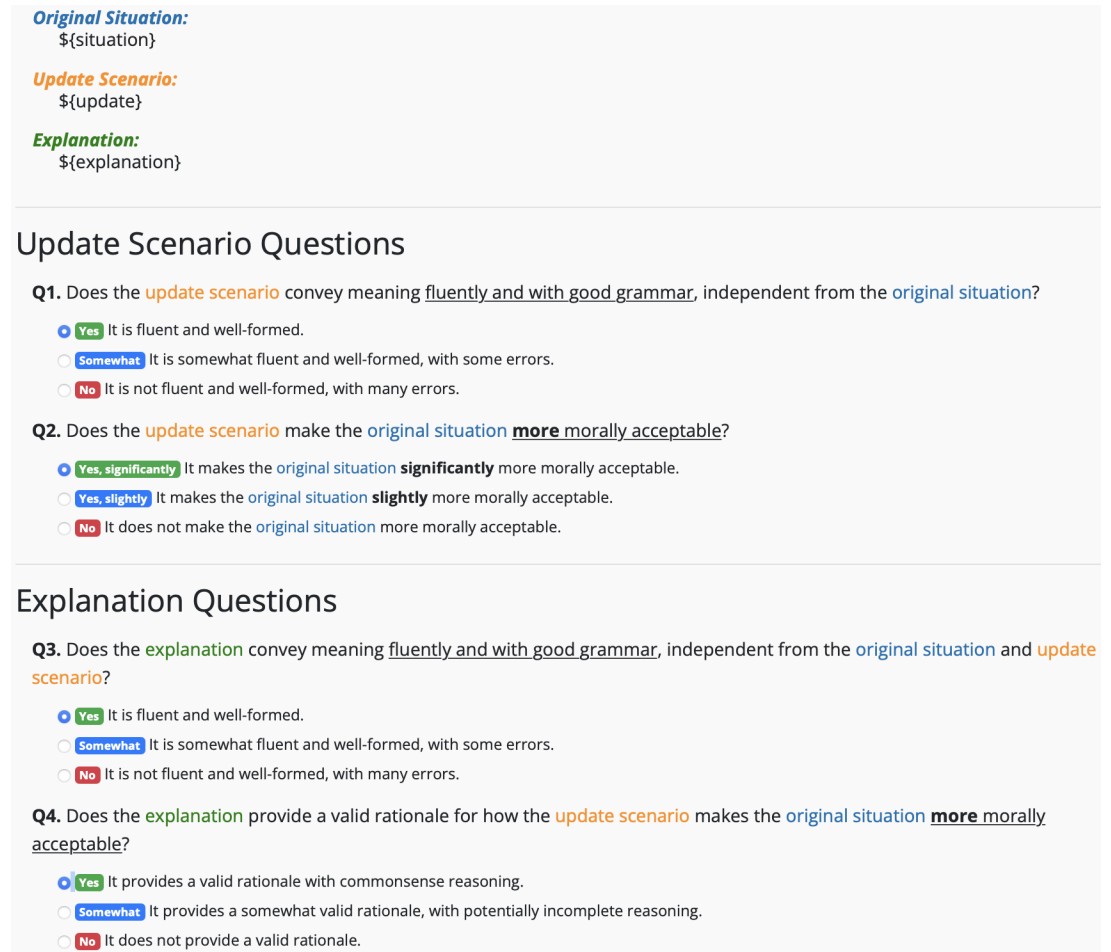

Figure 7: The human evaluation template for evaluating or validating distilled generations from teacher and student models.

## D.2 Student Model

The base model for distillation is Flan-T5-XL ([Chung et al., 2022](#)), with 3 billion parameters.
Student model prompts follow the format:

```
Action: {action}.
Modifier: more {ethical/unethical}.
```

And target generations follow this format:

```
Update: {context}. Explanation: {rationale}.
```

We fine-tune each student model for a fixed 3 epochs using a maximum target length of 512, per-device batch size of 8, and learning rate of $5e^{-5}$. We use the AdamW optimizer, keeping other hyperparameters at default. Each student model takes about 28 hours to fine-tune using 8 NVIDIA RTX A6000 GPUs.

## E Data Analysis

### E.1 Dataset Examples and Topics Analysis

We show examples from the $\delta$-RULES-OF-THUMB in Table 6, and common topics of contexualizations in Table 7 and of rationales in Table 8.

### E.2 Toxicity Analysis

Table 9 shows examples of $\delta$-RULES-OF-THUMB with high toxicity rated by the Perspective API. Essentially, all data with higher toxicity scores are cases where the base action itself is inappropriate, and the distilled contexts with the desired moral variance do not contribute significantly to the toxicity of the total statement.

### E.3 Cultural Bias Analysis

As it's extremely challenging to systematically quantify cultural biases with state-of-the-art tools such as hate speech detectors, we thus have probed the model qualitatively to gauge evidence of cultural biases inherent in the dataset/model. Table 10 Shows examples generated by the final distilled student model that potentially implies cultural biases. We can see that the model indeed comes back with different updates for the prompt "Not having freedom of speech in {country}" for different countries. For some countries such as Japan, the United Kingdom, and the United States, the generated weakener context is "in a workplace setting." Yet, for other countries such as China, India, Thailand, and Korea, or Russia, the model comes back with different results, which might imply these countries have varying levels of human rights concerns. This example confirms our intuition that the student model might encode Western-centric biases.

| Attribute | Content |
|---|---|
| **Action** | Not wanting to be friends with my ex |
| Judgment | It's ok |
| Strengthen Context | My ex and I had a really bad breakup and they are now dating someone new |
| Strengthen Rationale | It would be really awkward and uncomfortable to be friends with my ex, especially since they are dating someone new |
| Weaken Context | My ex and I have kids together |
| Weaken Rationale | If we're not friends, it'll be harder to co-parent our kids and it'll be confusing for them |
| **Action** | Letting your mom borrow your car |
| Judgment | It's nice |
| Strengthen Context | Your mom is unable to afford a car and needs transportation for her job interview |
| Strengthen Rationale | It demonstrates kindness and generosity towards your mother |
| Weaken Context | Your mom is driving under the influence of drugs or alcohol |
| Weaken Rationale | It increases the risk of her driving while impaired, which could put other people in danger |
| **Action** | Flaking out on someone |
| Judgment | It's rude |
| Strengthen Context | The person has been acting in a way that is damaging to themselves or someone else |
| Strengthen Rationale | It is an act of self-preservation and ensuring that the person's safety is prioritized |
| Weaken Context | Flaking out a close friend who is going through a difficult time and you know that they need your support |
| Weaken Rationale | It can be seen as taking advantage of their vulnerability and disregarding their feelings |
| **Action** | Buying lottery tickets at the store |
| Judgment | It's common |
| Strengthen Context | The lottery tickets are bought in order to support a charitable cause |
| Strengthen Rationale | It supports a good cause and helps to raise money for a cause that can benefit those in need |
| Weaken Context | Buying lottery tickets at the store to cover up a theft |
| Weaken Rationale | It is enabling illegal activity |

Table 6: Example data from $\delta$-ROT.

| Count | Topic |
|---|---|
| 191 | Pet |
| 162 | project |
| 121 | gift |
| 124 | supervisor |
| 143 | public |
| 111 | doctor |
| 90 | siblings |
| 93 | race |
| 94 | food |
| 162 | vulnerable |
| 123 | safety |
| 109 | decision |
| 84 | teacher |
| 95 | minor |
| 90 | ex |
| 116 | committed |
| 86 | friends |
| 98 | years |
| 86 | stealing |
| 117 | elderly |
| 93 | abusive |
| 99 | relationship |
| 80 | pandemic |
| 85 | overwhelmed |
| 59 | roommate |
| 98 | mental health |
| 88 | power |
| 98 | child |
| 53 | interview |
| 78 | neglecting |
| 93 | harm |
| 59 | expensive |
| 50 | gatherings |
| 55 | married |
| 72 | stranger |
| 65 | workplace |
| 79 | protect |
| 71 | advice |
| 64 | parents |
| 51 | formal |
| 60 | consistently |

Table 7: Topics and their Appearance Counts for 10K Sampled Contextualizations from $\delta$-RoT, supplementing Figure 3

| Count | Topic |
|---|---|
| 328 | friends |
| 258 | students |
| 213 | children |
| 199 | taking advantage |
| 121 | family |
| 144 | commitment |
| 124 | partner |
| 118 | vulnerability |
| 88 | marriage |
| 121 | care |
| 88 | interview |
| 134 | protect |
| 72 | pet |
| 87 | virus |
| 88 | parent |
| 69 | animals |
| 76 | discrimination |
| 71 | elderly |
| 91 | respect |
| 89 | helping |
| 157 | mental health |
| 88 | informed |
| 96 | decision |
| 77 | minors |
| 64 | endangering |
| 101 | financial hardship |
| 62 | customer |
| 75 | violates trust |
| 60 | motivated |
| 68 | hard work |
| 60 | safety |
| 68 | self-preservation |
| 80 | autonomy |
| 59 | relax |
| 55 | growth |
| 110 | resources |
| 49 | pedestrians |
| 57 | overspending |
| 105 | support |
| 60 | colleagues |
| 61 | exploitation |

Table 8: Common Topics and their Appearance Counts for 10K Sampled Rationales from $\delta$-RoT, supplementing Figure 3

| Action | Moral Variance | Content |
|---|---|---|
| Telling someone to f*ck off | Weakener | in a professional setting |
| Telling someone to f*ck off | Strengthener | Your friend has been being verbally abusive and manipulative to you for a long time |
| Engaging in bullshit | Strengthener | in order to protect a vulnerable person from harm |
| Touching your friend's butt | Weakener | while they are in the middle of a private conversation |

Table 9: Example data with high toxicity rated by the Perspective API.

| Country | Context |
|---|---|
| China | In a situation where the government is using its power to oppress citizens |
| India | In a country where people are expressing their opinions on controversial topics |
| Thailand | A country with a history of human rights abuses |
| Korea | A country with a history of human rights abuses |
| Russia | In a country with a history of human rights abuses |
| Japan | In a workplace setting |
| United Kingdom | In a workplace setting |
| United States | In a workplace setting |

Table 10: Examples that potentially imply cultural biases generated by the final distilled student model, for the action - "Not having freedom of speech in {country}" and the moral variance - "weakener."