# OpenReview forum: "What Makes it Ok to Set a Fire? Iterative Self-distillation of Contexts and Rationales for Disambiguating Defeasible Social and Moral Situations"
_EMNLP/2023/Conference — EMNLP 2023 Findings_

### Official Review · Reviewer_RfsH · 2023-07-26

**Soundness:** 3

**Excitement:**

3: Ambivalent: It has merits (e.g., it reports state-of-the-art results, the idea is nice), but there are key weaknesses (e.g., it describes incremental work), and it can significantly benefit from another round of revision. However, I won't object to accepting it if my co-reviewers champion it.

**Missing References:**

**1.** Kathleen C Fraser, Svetlana Kiritchenko, and Esma Balkir. “Does Moral Code Have a Moral Code? Probing Delphi’s Moral Philosophy

**2.** M. Abdulhai, S. Levine, and N. Jaques. “Moral Foundations of Large Language Models”. In: AAAI 2023 Workshop on Representation Learning for Responsible Human-Centric AI (2022).

**3.** Jochen Hartmann, Jasper Schwenzow, and Maximilian Witte. “The Political Ideology of Conversational AI: Converging Evidence on ChatGPT’s Pro-Environmental, Left-Libertarian Orientation”

**4.** Shibani Santurkar, Esin Durmus, Faisal Ladhak, Cinoo Lee, Percy Liang, and Tatsunori Hashimoto. “Whose Opinions Do Language Models Reflect?”

**5.** Julian Coda-Forno, Kristin Witte, Akshay K Jagadish, Marcel Binz, Zeynep Akata, and Eric Schulz. “Inducing Anxiety in Large Language Models Increases Exploration and Bias”



**Paper Topic And Main Contributions:**

The paper addresses the challenge of understanding how specific contexts influence moral judgments. Recognizing that moral decisions are often shaped by the nuances of their surrounding contexts, the authors introduce a task called "defeasible moral reasoning" to provide grounded contexts and rationales that determine the moral acceptability of actions. To gather data for this task, they employ an iterative self-distillation method, resulting in a high-quality dataset named δ-RULES-OF-THUMB (δ-ROT). This dataset, containing over a million entries, aids in training a model that captures the intricacies of human moral judgment in diverse real-life scenarios.

**Questions For The Authors:**

**A.** How did you ensure that the initial seed knowledge from GPT-3 was free from biases, and how did you mitigate the risk of these biases propagating through the iterative self-distillation process?

**B.** Can you elaborate on the generalizability of your methods? Specifically, how do you envision the iterative self-distillation approach and the δ-RULES-OF-THUMB (δ-ROT) dataset being adapted for other NLP tasks or diverse cultural contexts?

**C.** Given the sensitivity of moral reasoning and its cultural nuances, how did you account for potential cultural biases in the δ-ROT dataset? Were there any measures taken to ensure a diverse representation of moral perspectives?

**D.** What are the potential real-world applications you envision for the models trained on the δ-ROT dataset, and how do you plan to address the ethical implications associated with deploying such models?



**Reasons To Accept:**

The paper introduces a novel "defeasible moral reasoning" task, shedding light on how context influences moral judgments in NLP. Its main strength lies in the creation of the δ-RULES-OF-THUMB (δ-ROT) dataset, a comprehensive resource with over a million entries. The iterative self-distillation method used to generate this dataset is both interesting and scalable, with potential applications beyond this study.

**Reasons To Reject:**


1. **Over-reliance on iterative self-distillation**: The paper heavily depends on the iterative self-distillation method. If there are inherent biases or errors in the initial seed knowledge from GPT-3, they might propagate through the iterations, affecting the final dataset's quality.

2. **Dataset size over quality**: While the δ-RULES-OF-THUMB (δ-ROT) dataset is extensive, but it is unclear why is this better than a highly curated small dataset where examples are more representative of real-world moral dilemmas.

2. **Generalizability concerns**: The paper's focus on the specific task of "defeasible moral reasoning" might limit its generalizability. It's unclear how the proposed methods would perform on other NLP tasks or in different cultural and moral contexts.

3. **Potential ethical implications**: Moral reasoning is a complex and culturally sensitive area. There's a risk that the models and datasets might inadvertently perpetuate biases or oversimplify intricate moral dilemmas, leading to potential misuse in real-world applications.

4. **Lack of comparative analysis**: The paper might benefit from a more in-depth comparison with existing works or methods. Without this, it's challenging to gauge the true novelty or superiority of the proposed approach.

5. **Disregarding existing literature on morality and LLMs**: The paper fails to address the important and growing literature on the moral and social choices of LLMs. Please see missing references.



**Reproducibility:**

4: Could mostly reproduce the results, but there may be some variation because of sample variance or minor variations in their interpretation of the protocol or method.

**Reviewer Confidence:**

4: Quite sure. I tried to check the important points carefully. It's unlikely, though conceivable, that I missed something that should affect my ratings.

**Typos Grammar Style And Presentation Improvements:**

Figure 5 - “Example *of* rich contextualizations per category.”

Line 619 - comprising *of* over 1.2M combined

---

> ### Author Rebuttal · Authors · 2023-08-29
>
> We thank reviewer RfsH for their constructive suggestions! We are excited to see that they recognize our proposed task as “novel” and “shedding light on how context influences moral judgments in NLP” and recognize the value of our dataset and distillation pipeline! We address all their suggestions on the following points below:
>
> ### **1. Clarification of Over-reliance on Iterative Self-distillation and Method Generalizability**
>
> Thank you for raising the important question of whether our proposed iterative self-distillation method can be generalizable beyond the defeasible moral reasoning task and whether errors/biases from the teacher model will compound into student models.
>
> In a broader picture, our novel approach of iterative self-distillation is rooted in the rich line of research of symbolic knowledge distillation (i.e., distilling knowledge from teacher models to benefit the downstream student models), where **different variations of the distillation pipeline have been proven to be more effective than human annotations in a wide net of applications.** For example, commonsense knowledge [1], generic reasoning [2], chain-of-thought reasoning steps [3], summarization [4,5], paraphrasing [5], and dialogues [6]. Thus, we are confident that our iterative self-distillation approach has the potential to be highly flexible and adaptable to tasks beyond defeasible moral reasoning itself.
>
> > **The Iterative Self-distillation Method is Generalizable**
>
> Zooming into our iterative self-distillation approach, as it requires minimal task-specific constraints, it’s expected to be highly generalizable by nature. The approach's success comes from **three** main ingredients detailed below, and  **none of them are specifically bound to defeasible moral reasoning and thus could be generalizable to other reasonable, well-defined, open-text generative tasks.**
>
> (1) **An initial set of seed knowledge** that could be sourced using either automatic data generation or from existing datasets. In our case, we carefully prompted GPT-3 to generate the initial knowledge set. Still, we envision other variations of LLMs that can produce reasonable quality knowledge (e.g., Llama2) could also serve the purpose here.
>
> (2) **A critic model** that is trained on human annotations to mimic human ratings of the dataset's quality. Technically, for all tasks humans could evaluate, we can source human annotation data to train the critic model. Similar critic models have been proven to be effective in different applications in symbolic knowledge distillation research, such as commonsense knowledge [1] and generic reasoning knowledge [2].
>
> (3) **Iterative self-distillation and filtering** that amplifies desirable characteristics of the trained model. Our approach shows key advantages compared to previous symbolic knowledge distillation work. First, our approach minimizes the reliance on expensive API calls from closed-sourced models like GPT-3 as opposed to sourcing all data from GPT-3 [1,3]. Second, we use a simple over-generating and filtering pipeline instead of evoking complicated/slow constrained decodings during self-distillation [2]. We believe our simple but effective techniques could further lift the burden of operationalizing different tasks.
>
> > **The Critic Model Promotes Validity/Correctness of Knowledge Ported from the Teacher to the Student Models**
>
> Admittedly, the data quality from the teacher model is mediocre as-is without any treatment and thus will not serve as a good source for student models to train on (see Table 2 in the paper). However, with the careful screening of the critic filter and the NLI filter, we can increase the quality of intermediate student models via a richer set of training data from iterative self-distillation, and eventually get a final student model that produces much higher quality outputs rated by human evaluation (**Validity**: 0.56 -> 0.71, **Defeasibility**: 0.37 -> 0.56, **Language Quality**: 0.98 -> 0.99). To explicitly show the effectiveness of the critic model in promoting student model quality, we included ablation experiments to show the stark contrast between training student models for one iteration *with* vs. *without* filtering in Table 1 in the paper (**Validity**: 0.54 vs. 0.51, **Defeasibility**: 0.42 vs. 0.39). In the final version, we will clarify our careful data filtration process designed to counter the error propagation. Finally, we acknowledge that even with our filtering by the critic model, hallucinated outputs can still be produced by distilled models due to the limitations of critic models (see discussion in Section 8). We will stress in the final version that users must exercise discretion and care when applying our dataset to novel use cases.
>
> - [1] Symbolic Knowledge Distillation: from General Language Models to Commonsense Models: https://arxiv.org/abs/2110.07178
> - [2] I2D2: Inductive Knowledge Distillation with NeuroLogic and Self-Imitation: https://arxiv.org/abs/2212.09246
> - [3] Symbolic Chain-of-Thought Distillation: Small Models Can Also “Think” Step-by-Step: https://arxiv.org/abs/2306.14050
> - [4] Referee: Reference-Free Sentence Summarization with Sharper Controllability through Symbolic Knowledge Distillation: https://arxiv.org/abs/2210.13800
> - [5] Impossible Distillation: from Low-Quality Model to High-Quality Dataset & Model for Summarization and Paraphrasing: https://arxiv.org/abs/2305.16635
> - [6] SODA: Million-scale Dialogue Distillation with Social Commonsense Contextualization: https://arxiv.org/abs/2212.10465
>
> ### **2. Dataset Size and Quality**
>
> We appreciate the concern about dataset quality—high-quality data is indeed important, and increasing dataset size at the expense of quality would arguably decrease the value of the dataset. To counter this trend, to produce δ-ROT, we set a high critic threshold of 0.96 to maintain high quality in the data (Section 3.3). This threshold removes 73% of the candidate generations to produce the restricted set of generations in δ-ROT. Human annotators deem 85.9% of the contextualizations and 99.8% of the rationales as high-quality (Section 2). This is on par with similar datasets in the field [1], and using the critic model to enforce quality in the data is much more efficient than human annotations, which could be very prohibitive when scaling up. Thus, it’s fair to say that our dataset is both large-scale and high-quality.
>
> We also note that scale is an important factor in the value of the dataset to train models. In our own data, we can compare SelfDistill1 to the ablation with Top 1 Only, which has high data quality yet smaller size due to the greedy sampling. In this ablation, we observe that SelfDistill1, with a larger scale training set, performs better on the number of human-judged valid generations, even when ignoring the uniqueness constraint (Table 2). In other literature, it is also observed that language model performance scales with dataset size according to the LM scaling law, demonstrating that both dataset scale and quality are critical to producing high-quality models [2].
>
> It’s also a valid concern to ensure that the dataset reflects real-world moral dilemmas. Hence, we source the actions in δ-ROT from Social Chemistry 101, which is filled with a “rich spectrum of real-life situations” that carry sociomoral norms and judgments [3]. By building our dataset on top of these real-world actions, δ-ROT is representative of moral contextualizations that people are likely to encounter in the real world.
>
> - [1] Symbolic Knowledge Distillation: from General Language Models to Commonsense Models: https://arxiv.org/abs/2110.07178
> - [2] Scaling Laws for Neural Language Models: https://arxiv.org/abs/2001.08361
> - [3] Social Chemistry 101: Learning to Reason about Social and Moral Norms: https://arxiv.org/abs/2011.00620
>
> ### **3. Discussion on Potential Biases**
>
> Thank you for calling our attention to discuss the important point of potential biases of our dataset. We sourced the seed actions from Social Chemistry 101 [1] that concern mostly everyday situations, such as pet, project, gift, supervisor, public, and siblings (see details of the topic analysis of our dataset in Table 6/7 in the appendix). Thus, we find them less prone to biases/toxicity to begin with. In addition, due to the careful filtering with the critic model, and the iteratively refined self-distillation process, we expect most of the low-quality (including potentially biased data points) are already filtered out from the final dataset. This is also why human annotations deem our final distilled dataset high quality overall.
>
> However, we acknowledge the importance of examining the potential biases, especially considering the sensitive nature of morality and the potentially limited scope of the critic model. Therefore, we run additional experiments of toxicity analysis with the Perspective API on a subset of 40K data points of our final dataset. Our results show that the average toxicity score is **0.09**, indicating very low toxicity overall. In a qualitative analysis of the data rows with higher toxicity scores (with a max of *0.83*), essentially all of them are cases where the base action itself is inappropriate/explicit, and the distilled contexts that have the desired moral variance do not contribute significantly to the toxicity of the total statement. *We provide some examples with high toxicity scores in the Table below.* While *no* existing toxicity detection can accurately measure all potential biases, we will incorporate Perspective API analysis and thoroughly discuss potential limitations of this toxicity analysis.
>
> | **Action**                      | **Moral Variance** | **Context**                                                                         |
> |---------------------------------|--------------------|-------------------------------------------------------------------------------------|
> | Telling someone to f*ck off     | Weakener           | in a professional setting                                                           |
> | Engaging in bullshit            | Strengthener       | in order to protect a vulnerable person from harm                                   |
> | Touching your friend's butt     | Weakener           | while they are in the middle of a private conversation                              |
> | Telling your friend to f*ck off | Strengthener       | Your friend has been being verbally abusive and manipulative to you for a long time |
>
> Finally, we want to highlight that our dataset should **never** be used as prescriptive advice to humans. Our work aims to promote the consideration of defeasible contexts to state underspecified contexts in moral reasoning explicitly. This opens future avenues for more nuanced interpretation/consideration in computational morality research. We will add this important point to the final paper as well.
>
> - [1] Social Chemistry 101: Learning to Reason about Social and Moral Norms: https://arxiv.org/abs/2011.00620
>
> ### **4. Discussion about Accommodating Diverse Perspectives and Cultural Nuances**
>
> We appreciate the reviewer for raising the important questions of how we account for diverse representations of moral perspectives and potential cultural biases, which we underspecified in our original draft. We will discuss this point in the following response and will add them to the camera ready.
>
> > **Diversity**
>
> We readily agree with the reviewer that “moral reasoning is a complex and culturally sensitive area” and thus, it is important to accommodate various different but equally valid opinions for making judgments. Our work is precisely motivated by this observation: *oftentimes, the failure to accommodate variations of moral judgments is due to the underspecified and unspoken assumptions or contexts of the given default action.* For example, the default/average judgment of the action “talking loudly” will be shifted in various directions with more detailed context specifications (e.g., if you try to warn people during a fire, in a public library). **To this goal, we introduced the defeasible moral reasoning task to explicitly gauge various rich defeasible contextualizations that underpin alternative perspectives and shifts of moral judgments in nuanced but essential ways.**
>
> With this goal in mind, we took precautions to accommodate/promote diversity of representations during data distillation. **First**, we don’t presume any default moral judgment given an action (i.e., if the action is *absolutely* good or bad) but rather to understand various contexts that shift the relative moral implications of a situation differently (i.e., whether a context makes the given action *MORE* or *LESS* morally acceptable). This assumption is helpful to not force default/average interpretation or frame onto an action, but to attempt to justify the rich variability of its judgments based on alternative contexts and interpretations. **Second**, it is important to note that an action could be contextualized in many ways. Thus, we deliberately promote the distillation of multiple contextualizations to account for diverse ways of defeating moral judgments. As a result, we improved the diversity scores significantly throughout the self-distillation process (4.78 -> 5.69, see Table 2 in the paper). Therefore, although we do not directly engage different moral perspectives during distillation, such as philosophical moral theories, we contribute to understanding various perspectives that moral judgments could be shifted due to variable contexts, complementing the existing landscape of computational morality research.
>
> > **Culture Sensitivity and Nuances**
>
> Among all perspectives that shape people’s morality differently, one crucial dimension of variation is people’s cultural background. As we responded to ethical implications and general biases, our final dataset exhibited *very low* indications of overall toxicity. However, it is hard to conclude that our dataset/model is immune from cultural biases, as even the best hate speech detector to date might fail to capture this more nuanced type of bias. In fact, previous studies have pointed out that cultural bias is a pervasive phenomenon across many NLP models (e.g., GPT-3/3.5/4) and tasks (e.g., hate speech detection with Perspective API, RewireAPI, HateRoberta) [7]. Admittedly, as our dataset and student models are distilled from GPT-3, which is shown to present *Western-centric* perspectives, it is highly likely that our dataset and models inherit this cultural bias as well.
>
> As it’s extremely challenging to systematically quantify cultural biases with state-of-the-art tools such as hate speech detectors, we thus have probed the model qualitatively to gauge evidence of cultural biases inherent in the dataset/model. We provide some interesting examples in the Table below. We can see that the model indeed comes back with different updates for the prompt “Not having freedom of speech in {country}” for different countries. For some countries such as Japan, the United Kingdom, and the United States, the generated weakener context is “in a workplace setting.” Yet, for other countries such as China, India, Thailand, and Korea, or Russia, the model comes back with different results, which might imply these countries have varying levels of human rights concerns. This example confirms our intuition that the student model might encode Western-centric biases.
>
> **Action Template**: “Not having freedom of speech in {country}”
>
> **Moral Variance**: “weakener”
>
> | **Country**    | **Context**                                                                     |
> |----------------|---------------------------------------------------------------------------------|
> | China          | In a situation where the government is using its power to oppress citizens      |
> | India          | In a country where people are expressing their opinions on controversial topics |
> | Thailand       | A country with a history of human rights abuses                                 |
> | Korea          | A country with a history of human rights abuses                                 |
> | Russia         | In a country with a history of human rights abuses                              |
> | Japan          | In a workplace setting                                                          |
> | United Kingdom | In a workplace setting                                                          |
> | United States  | In a workplace setting                                                          |
>
> We will add the full probing results to the paper and an extensive discussion of the potential Western-centric cultural biases due to distillation to the final paper. Further, we will deepen the discussion of future directions to lay out concrete road maps of future research on enriching cultural representations in computational moral reasoning research. Finally, we will make our data and code completely open source to allow for scrutiny and improvement by the research community.
>
> Finally, it's worth noting that discussions on the feasibility of instilling morally-informed AI are often marked by deep-seated disagreements concerning the influence of culture, personal experiences, and specific circumstances on people's moral perspectives. These disparities have led some to assert that endeavoring to systematically imbue machines with human moral values is futile. However, evidence spanning various disciplines, including comparative law [1], anthropology [2], psychology [3,4,5], and philosophy [6], challenges this notion by suggesting that fundamental similarities underlie human moral systems, potentially outweighing their divergences. **In light of this, it’s equally, if not more, important to start investigating the possibility of teaching models to capture matters of widespread moral consensus under defeasible variations.** In particular, such differences do not necessarily preclude the need for investigating cases where people tend to agree. To this end, we will add the discussion of the importance of moral consensus in addition to moral disagreement into the paper.
>
> - [1] John Mikhail. “Is the Prohibition of Homicide Universal-Evidence from Comparative Criminal Law”. In: Brook. L. Rev. 75 (2009), p. 497.
> - [2] Oliver Scott Curry, Daniel Austin Mullins, and Harvey Whitehouse. “Is it good to cooperate? Testing the theory of morality-as-cooperation in 60 societies”. In: Current Anthropology 60.1 (2019), pp. 47–69.
> - [3] PR Blake et al. “The ontogeny of fairness in seven societies”. In: Nature 528.7581 (2015), pp. 258–261.
> - [4] Edmond Awad et al. “Universals and variations in moral decisions made in 42 countries by 70,000 participants”. In: Proceedings of the National Academy of Sciences 117.5 (2020), pp. 2332–2337.
> - [5] H Clark Barrett and Rebecca R Saxe. “Are some cultures more mind-minded in their moral judgements than others?” In: Philosophical Transactions of the Royal Society B 376.1838 (2021), p. 20200288.
> - [6] Bernard Gert. “Common Morality and Computing”. In: Ethics and Information Technology 1.1 (1999), pp. 53–60. doi: 10.1023/a:1010026827934.
> - [7] Sebastin Santy et al. “NLPositionality: Characterizing Design Biases of Datasets and Models”: https://arxiv.org/abs/2306.01943
>
> ### **5. Comparative Analysis**
>
> Our current work tackles the novel and unique task of defeasible moral reasoning. To the best of our knowledge, no existing work tackles this exact task, and thus, it's challenging to find other comparative task-specific baselines. The closest related works that we had discussed in the paper introduction are [1] (on clarification question generation for moral situations) and [2] (simple situational norms), but none of these provide outcome models that directly produce defeasible moral contexts and rationales compatible with our task.
>
> Instead, we conducted new experiments to investigate the zero-shot performance of some of the most powerful general-purpose LLMs of varied sizes with 1000 sampled examples from our test set. We use the same instructions as we used to distill the seed knowledge from GPT-3 to prompt these baselines in a zero-shot manner (see Section A in the paper's appendix). We evaluate them using the comparable automatic evaluation metric via the critic model (i.e., the same metrics used in Table 1 in the paper). Such automatic evaluation scores are shown to be highly correlated with human evaluation results in our paper. **Our results show that despite some of the zero-shot models being orders of magnitude larger than our final student model (e.g., 175B vs. 3B), our model outperforms them all, proving the effectiveness of our proposed approach.** It's also worth noting that our models will also be open-sourced, providing further accessibility advantages compared to some of these massive and opaque closed-sourced LLMs. We will add a detailed discussion of these new baselines to the final paper!
>
> | **Model**          | **Percentage of Valid Contexts**     | **Avg. Critic Score**     |
> |--------------------|--------------|--------------|
> | *(NEW)* Falcon-7B-Instruct | 0.39		 | 0.54 		 |
> | GPT-3              | 0.53         | 0.69         |
> | *(NEW)* GPT-3.5 (ChatGPT)  | 0.71         | 0.77         |
> | *(NEW)* GPT-4              | 0.77         | 0.82         |
> | **Ours**           | **0.86**     | **0.88**     |
>
> - [1] ClarifyDelphi: Reinforced Clarification Questions with Defeasibility Rewards for Social and Moral Situations: https://arxiv.org/abs/2212.10409
> - [2] NormBank: A Knowledge Bank of Situational Social Norms: https://arxiv.org/abs/2305.17008
>
> ### **6. Potential Real-world Application and Strategies for Addressing Ethical Implications**
> > **Potential Real-world Applications**
>
> Here’s a list of potential use cases of our task/dataset. This is by no means a complete list, but we hope it could provide a fuller picture of the downstream potential of our work!
> - Incorporating the nuanced understanding of defeasible context into reasoning about social norms and values in interactive narrative games, e.g., [1,2,3]
> - Using the richer set of defeasible contexts to reward more interesting clarification question generation for disambiguating social and moral situations, e.g., [4]
> - Using the contextualized understanding of ethical situations to enhance prosocial conversation in dialogue applications, e.g., [5]
>
> In the broader scope, it’s critical to understand that the defeasible, contextualized nuances of moral reasoning itself is largely understudied in the current landscape of computational morality research, and thus our task and dataset inspires further research in understanding the interleaving complexity of moral reasoning. Finally, as we mentioned in the method generalization response, we believe our proposed iterative self-distillation method serves as a simple but effective method for other tasks.
>
> > **Strategies for Addressing Ethical Implications**
>
> We will publicly release our dataset and final student model to promote open-source research but for ​​gated research purposes only. To mitigate risks of misuse, we will inform all users about potential ethical implications and risks of the resource and require them to complete a data/model user agreement form to acknowledge their consent to proper usage of the resource. This will also help us track how our resource is repurposed for downstream applications. We will also add explicit discussions to encourage future research to tackle limitations of our current work, such as potential cultural biases. We'll add the above discussion to the final paper.
>
> - [1] What Would Jiminy Cricket Do? Towards Agents That Behave Morally: https://arxiv.org/abs/2110.13136
> - [2] Aligning to Social Norms and Values in Interactive Narratives: https://arxiv.org/abs/2205.01975
> - [3] Do the Rewards Justify the Means? Measuring Trade-Offs Between Rewards and Ethical Behavior in the MACHIAVELLI Benchmark: https://arxiv.org/abs/2304.03279
> - [4] ClarifyDelphi: Reinforced Clarification Questions with Defeasibility Rewards for Social and Moral Situations: https://arxiv.org/abs/2212.10409
> - [5] ProsocialDialog: A Prosocial Backbone for Conversational Agents: https://arxiv.org/abs/2205.12688
>
> ### **7. More Extensive Discussion of the Literature on Morality and LLMs**
>
> Thank you for pointing out that we should expand our discussion on computational morality and LLMs! We originally discussed computational morality in the related work section, and we appreciate the additional literature you suggested. We’ll include them all in the camera-ready to address related work more comprehensively!
>
> ### **8. Typos and Presentation Improvements**
> Thank you for pointing out the typos; we’ll fix them all in the camera-ready draft!

---

### Official Review · Reviewer_7Yd4 · 2023-08-03

**Soundness:** 4

**Excitement:**

3: Ambivalent: It has merits (e.g., it reports state-of-the-art results, the idea is nice), but there are key weaknesses (e.g., it describes incremental work), and it can significantly benefit from another round of revision. However, I won't object to accepting it if my co-reviewers champion it.

**Missing References:**

Kim, H., Yu, Y., Jiang, L., Lu, X., Khashabi, D., Kim, G., ... & Sap, M. (2022). Prosocialdialog: A prosocial backbone for conversational agents. arXiv preprint arXiv:2205.12688.

**Paper Topic And Main Contributions:**

The authors propose a new task of defeasible moral reasoning where an action is associated with its moral connotations based on the context. They curate a dataset to support this task with 1.2M instances. They use distinct models as teacher and student models to obtain the required annotations and perform human evaluation to validate them.

**Reasons To Accept:**

- Interesting task definition and clear reason for it to be useful in daily applications
- Large dataset introduced that can be used by other researchers working in a similar field.
- Extensive detail about the proposed dataset and the curation pipeline.
- Good diagrams with detailed captions.

**Reasons To Reject:**

- A bit hard to follow. Writing can be improved. Also, can bring the related work section after introduction so that readers can get accustomed to the content of the paper.
- The paper seems like a complete data paper. A preliminary study on the proposed dataset would have been highly beneficial.
- The dataset introduced is extremely similar to the ProsocialDialogue dataset [1]. However, it is not cited anywhere.

[1] Kim, H., Yu, Y., Jiang, L., Lu, X., Khashabi, D., Kim, G., ... & Sap, M. (2022). Prosocialdialog: A prosocial backbone for conversational agents. arXiv preprint arXiv:2205.12688.

**Reproducibility:**

4: Could mostly reproduce the results, but there may be some variation because of sample variance or minor variations in their interpretation of the protocol or method.

**Reviewer Confidence:**

3: Pretty sure, but there's a chance I missed something. Although I have a good feel for this area in general, I did not carefully check the paper's details, e.g., the math, experimental design, or novelty.

---

> ### Author Rebuttal · Authors · 2023-08-29
>
> We thank reviewer 7Yd4 for their endorsement of acceptance and valuable feedback! We are excited that they recognize our proposed task as “interesting” and “useful in daily applications” and recognize our proposed dataset and curation pipeline as having extensive details that could be useful for future related research! We address their specific comments on the following points:
>
> ### **1. Writing Improvement**
>
> Thank you for pointing this out; we will do another pass over the paper to revise the writing quality and flow for the camera-ready version and move the related work section to follow the introduction per your suggestion.
>
> ### **2. Clarification of Paper Contribution and Data Analysis**
>
> We agree that a more in-depth study of the dataset would be a fitting direction to continue this line of work. Recognizing the importance of it, we have done a preliminary dataset analysis, presented in Section 5.2 (see Fig. 4 & 5). We present insights over topics represented in δ-ROT surfaced through both modeling and qualitative analysis, categories of contextualizations and rationales broken into each moral variance, and the most common failure modes in the second-iteration distilled model. Our preliminary analysis reveals interesting insights on contextualizations that vary the degree of moral acceptability and underlying rationales that justify moral decisions. The results from this analysis include that vulnerability of the subject of an action is a critical weakening context and that (in)equity or (un)fairness in practice carry significant weight on moral implications; these observations also align with theories in cognitive science literature [1].
>
> In addition, we would like to clarify that our contributions go beyond the dataset. Our work also includes the introduction of the defeasible moral reasoning task and a novel self-distillation pipeline, which we empirically show to improve generation quality and diversity through automated metrics and human evaluations of generations from our trained model.
>
> - [1] The Theory of Dyadic Morality: Reinventing Moral Judgment by Redefining Harm: https://journals.sagepub.com/doi/full/10.1177/1088868317698288
>
> ### **3. Distinction from ProsocialDialog**
>
> Thank you for pointing out this missing reference! We want to highlight some key differences between our work and ProsocialDialog. Our work introduces the task of **defeasible moral reasoning**, which reasons about grounded contexts that make a given action more or less morally acceptable, along with rationales based on implicit social and cultural norms. Yet, ProsocialDialog introduces a **multi-turn dialogue dataset** that teaches dialogue agents to respond to problematic content following social norms. Although the motivation of both datasets is guided by an understanding of social and cultural norms, they are significantly different in terms of both the goal (*ours*: defeasible moral reasoning vs. *ProsocialDialog*: dialogue safety) and the form (*ours*: open-text generation of defeasible contexts and rationales vs. *ProsocialDialog*: dialogues). We will add ProsocialDialog as a reference and clarify this distinction in the final draft!

---

### Official Review · Reviewer_Uk2j · 2023-08-05

**Soundness:** 4

**Excitement:**

5: Transformative: This paper is likely to change its subfield or computational linguistics broadly. It should be considered for a best paper award. This paper changes the current understanding of some phenomenon, shows a widely held practice to be erroneous in someway, enables a promising direction of research for a (broad or narrow) topic, or creates an exciting new technique.

**Paper Topic And Main Contributions:**

This paper introduces the task of defeasible moral reasoning, which involves providing contexts that alter the moral acceptability of an action, along with rationales that explain the reasoning. The main contributions are:

- Formulation of the defeasible moral reasoning task.
- Introduction of the δ-RULES-OF-THUMB (δ-ROT) dataset containing 1.2M contexts and rationales for 115K actions.
- An iterative self-distillation methodology to create high-quality training data and models for the task.
- Analysis of the dataset contents and model outputs, showing improved diversity and validity.

**Questions For The Authors:**

In the Human Critic Gold Data section you used the Majority Rule to aggregate labels, Is there any reason? See:
Kruger, J., Endriss, U., Fernández, R., & Qing, C. (2014). Axiomatic analysis of aggregation methods for collective annotation.

**Reasons To Accept:**

- The defeasible moral reasoning task is novel and challenging.
- Introduction of the δ-ROT dataset. The dataset size and quality metrics suggest δ-ROT will be a useful resource for future work.
- The iterative distillation process is effective at improving model quality over raw GPT-3 generations. Using this approach to move from expensive API calls to smaller models is valuable.
- This paper is detailed and easy to follow with clear logic.


**Reasons To Reject:**

- We know that a frequent problem with using large language models is hallucinations, even if we carefully control the model. In the teacher -student model presented in this thesis, I think there would also be teacher-misleading student behavior, and I hope that a corresponding discussion will appear in the final version.

**Reproducibility:**

3: Could reproduce the results with some difficulty. The settings of parameters are underspecified or subjectively determined; the training/evaluation data are not widely available.

**Reviewer Confidence:**

4: Quite sure. I tried to check the important points carefully. It's unlikely, though conceivable, that I missed something that should affect my ratings.

---

> ### Author Rebuttal · Authors · 2023-08-29
>
> We thank reviewer Uk2j for their endorsement of acceptance and constructive feedback! We are thrilled to see that they recognize our proposed task as “novel and challenging”, our contributed dataset as a “useful resource for future work”, our distillation process as “valuable”, and rate our paper as highly exciting! Below, we address specific comments they made:
>
> ### **1. How hallucination from the teacher model will impact the student model?**
>
> Thank you for calling our attention to the important point of hallucination of large language models. You’re correct that as our distillation framework ports knowledge from the large teacher model to smaller student models, it’s likely that the smaller models may also inherit undesirable behaviors. However, as we employ the critic model that’s trained with human annotations during the iterative self-distillation process, low-quality teacher model generations, including potential hallucinations such as logical inconsistencies and factual errors, are filtered out by the critic model, and thus might not present in the successive rounds of training of student models. As a result, we are able to increase the quality of intermediate student models via a richer set of training data from iterative self-distillation, and eventually get a final student model that produces much higher quality outputs based on human evaluation (**Validity**: 0.56 -> 0.71, **Defeasibility**: 0.37 -> 0.56, **Language Quality**: 0.98 -> 0.99). To explicitly show the effectiveness of the critic model in promoting student model quality, we included ablation experiments to show the contrast between training student models *with* vs. *without* filtering in Table 1 in the paper (**Validity**: 0.54 vs. 0.51, **Defeasibility**: 0.42 vs. 0.39). In the final version, we will clarify our careful data filtration process designed to counter the error propagation due to hallucination. Finally, we acknowledge that even with our filtering by the critic model, hallucinated outputs can still be produced by distilled models due to the limitations of critic models (see discussion in Section 8). We will stress in the final version that users must exercise discretion and care when applying our dataset to novel use cases.
>
> ### **2. Clarification on the Human Critic Data Aggregation Strategy**
>
> Thank you for providing this reference and calling out the possibility of different aggregation strategies! The work on bias-correcting rules is quite interesting. We used a majority rule for simplicity because there are only two classes (valid/invalid) and three annotators, and since previous work that similarly used crowdsourcing for commonsense-domain critic data also used a majority rule [1].
>
> However, when releasing the final dataset along with the camera-ready paper, we plan to include the individual gold annotations along with the aggregated results so that downstream users of the dataset can choose any preferred aggregation strategy. We agree that exploring additional aggregation strategies could be a promising direction to seek improvements to the distillation process in future work!
>
> - [1] Symbolic Knowledge Distillation: from General Language Models to Commonsense Models: https://arxiv.org/abs/2110.07178

---

### Meta-Review · Area_Chair_doL9 · 2023-09-19

**Recommendation:** 5

**Metareview:**

The paper presents a new task, defeasible moral reasoning, to provide grounded contexts that make an action more or less morally acceptable, along with commonsense rationales that justify the reasoning.

The authors introduced the δ-RULES-OF-THUMB (δ-ROT) dataset of 1.2M entries of combined context and rationales for 115K defeasible moral actions. δ-ROT is created through an iterative approach, which includes self-distillation, targeted filtering using a critic model trained with human judgment and (3) self-imitation learning.


Pros:

- The δ-ROT dataset is an important contribution with over 1.2M entries.  It also contains rationales that *explain the moral judgment of an action* that is useful for the community.
- The method to generate the data is not novel but effective and can be used for other tasks.
- The qualitative analysis of contextualised categories per moral variance is interesting.

Cons:
- Strong baselines such as finetuned open-sourced models are missing.

In the result section, there are too many **bold** words. Try to use italics instead.

Finally, The AreaChair appreciate the authors' response to clarify the doubts of the reviewers.

---

### Decision · Program_Chairs · 2023-10-07

**Decision:**

Accept-Findings

**Comment:**

The paper presents a new task, defeasible moral reasoning, to provide grounded contexts that make an action more or less morally acceptable, along with commonsense rationales that justify the reasoning.

The authors introduced the δ-RULES-OF-THUMB (δ-ROT) dataset of 1.2M entries of combined context and rationales for 115K defeasible moral actions. δ-ROT is created through an iterative approach, which includes self-distillation, targeted filtering using a critic model trained with human judgment and (3) self-imitation learning.


Pros:

- The δ-ROT dataset is an important contribution with over 1.2M entries.  It also contains rationales that *explain the moral judgment of an action* that is useful for the community.
- The method to generate the data is not novel but effective and can be used for other tasks.
- The qualitative analysis of contextualised categories per moral variance is interesting.

Cons:
- Strong baselines such as finetuned open-sourced models are missing.

In the result section, there are too many **bold** words. Try to use italics instead.

Finally, The AreaChair appreciate the authors' response to clarify the doubts of the reviewers.